# Array Radar Three-Dimensional Forward-Looking Imaging Algorithm Based on Two-Dimensional Super-Resolution [note 1]

**DOI:** 10.3390/s24227356

**Published:** 2024-11-18

**Authors:** Jinke Dai, Weijie Sun, Xinrui Jiang, Di Wu

**Affiliations:** 1The Key Laboratory of Radar Imaging and Microwave Photonics, Nanjing University of Aeronautics and Astronautics, Nanjing 211106, China; 1449715852@nuaa.edu.cn (J.D.); 952193955@nuaa.edu.cn (W.S.); 2Shenzhen Research Institute, Nanjing University of Aeronautics and Astronautics, Shenzhen 518038, China; 3Beijing Institute of Remote Sensing Equipment, Beijing 100854, China; jane_nudt@163.com

**Keywords:** array radar, forward-looking, super-resolution technique, three-dimensional imaging

## Abstract

Radar imaging is a technology that uses radar systems to generate target images. It transmits radio waves, receives the signal reflected back by the target, and realizes imaging by analyzing the target’s position, shape, and motion information. The three-dimensional (3D) forward-looking imaging of missile-borne radar is a branch of radar imaging. However, owing to the limitation of antenna aperture, the imaging resolution of real aperture radar is restricted. By implementing the super-resolution techniques in array signal processing into missile-borne radar 3D forward-looking imaging, the resolution can be further improved. In this paper, a 3D forward-looking imaging algorithm based on the two-dimensional (2D) super-resolution algorithm is proposed for missile-borne planar array radars. In the proposed algorithm, a forward-looking planar array with scanning beams is considered, and each range-pulse cell in the received data is processed one by one using a 2D super-resolution method with the error function constructed according to the weighted least squares (WLS) criterion to generate a group of 2D spectra in the azimuth-pitch domain. Considering the lack of training samples, the super-resolution spectrum of each range-pulse cell is estimated via adaptive iteration processing only with one sample, i.e., the cell under process. After that, all the 2D super-resolution spectra in azimuth-pitch are accumulated according to the changes in instantaneous beam centers of the beam scanning. As is verified by simulation results, the proposed algorithm outperforms the real aperture imaging method in terms of azimuth-pitch resolution and can obtain 3D forward-looking images that are of a higher quality.

## 1. Introduction

MISSILE-BORNE radar has become a key area of research for many countries. This radar system is an integral part of missile technology, providing precise target guidance and detection lock capabilities. It holds significant strategic importance in modern warfare. So three-dimensional (3D) imaging of missile-borne radar has been paid great attention to by many researchers. Currently, a number of 3D imaging methods for synthetic aperture radar (SAR), such as interferometric SAR (InSAR), tomography SAR, sparse array SAR and curvilinear SAR [1,2,3,4], have been tested via groups of real radar data. However, these methods only work in the side-looking and squint-looking modes and are invalid in the forward-looking region. This is because SAR requires large Doppler bandwidth of the observed scene to obtain a high-resolution in azimuth, which is hard to achieve in the forward-looking region due to the rapid decrease in the Doppler gradient [5].

The three-dimensional forward-looking imaging technique plays a significant role in missile guidance, autonomous helicopter landing, and urban mapping. Currently, forward-looking SAR (Flo-SAR) is a relatively popular radar system in 3D forward-looking imaging. In this system, the linear array is placed in the cross-track direction. The array elements transmit and receive sequentially and are equivalently regarded as the motion of radar in the cross-track direction. A. Reigber first applied it to 3D forward-looking imaging and completed a series of simulations [6]. To further improve the resolution of Flo-SAR, R. Yang proposed a backpropagation algorithm to improve azimuth-pitch resolution [7]. However, the computational complexity of the proposed method is huge. In [8], the sparse sampling theory was introduced into the Flo-SAR system, which effectively reduces the processing time of 3D reconstruction. In [9], the monopulse technique was implemented in 3D forward-looking imaging. With this technique, the imaging accuracy and imaging speed are significantly improved.

The above are some 3D forward-view imaging methods proposed by predecessors. Owing to the limitation of the physical aperture of the antenna, improvement of the resolution is limited. Another feasible approach to further improve the resolution of forward-looking radar imaging is to employ the array radar system and introduce the super-resolution technique into the procedure of imaging. Recently, array super-resolution techniques have been applied to forward-looking radar imaging, and the performance has been proved by groups of real data [10,11,12]. In this paper, we deal with the 3D forward-looking imaging problem by adopting a two-dimensional (2D) array radar (planar array). In addition, we focus on the 2D super-resolution algorithm to improve the resolution of 3D imaging.

As we know, a number of mature super-resolution algorithms can be employed in array processing, such as the 2D Multiple Signal Classification (MUSIC) and 2D CAPON algorithms. However, these algorithms require a considerable accumulation of snapshots to maintain estimation accuracy, which are usually obtained from the surrounding range gates of the cell under test in practical processing [13]. This means that this method is unsuitable for radar imaging when a super-resolution algorithm is employed, as each range gate of the echo data corresponds to a different scene.

In this paper, the super-resolution method, which works under the condition of fewer-sample or even single-sample support, is mainly considered. In [14], the 2D iterative adaptive approach (2D-IAA) was used in a stationary real aperture radar platform, and the azimuth-pitch 2D reconstruction was obtained. The advantage of IAA is the angle super-resolution that can be obtained with a single snapshot [15]. Inspired by [14], we apply the 2D-IAA to the missile-borne radar and propose an array radar 3D forward-looking imaging algorithm based on 2D super-resolution. In this algorithm, array radar scans along the azimuth. The single transmitting antenna transmits coherent pulses, and the planar array receives echoes. After the pulse compression processing, we apply 2D-IAA to process the echo data of each range-pulse cell. A series of 2D super-resolution spectra can be obtained. Then, the 2D super-resolution spectra in the same range cell are accumulated according to the changes in instantaneous beam centers of the beam scanning. The azimuth-pitch super-resolution can be obtained. Finally, the 3D forward-looking imaging results in the space rectangular coordinate system are obtained with coordinate transformation. The simulation results manifest that the 3D imaging resolution is significantly improved.

This paper is organized as follows. In Section 2, the azimuth-pitch echo model of array radar is established. In Section 3, the principle of 2D-IAA and the processing flow of 3D forward-looking imaging algorithm are discussed in detail. In Section 4, simulations are presented to verify the validity of the method. In Section 5, research conclusions are given.

## 2. Azimuth-Pitch Echo Model

A three-dimensional (3D) geometry in a Cartesian coordinate system is shown in Figure 1, where a planar array mounted on a motion platform flying along the y-direction is considered. The total number of the independent receiving channels of the array is $M_{1}\times M_{2}$, where the number of channels along the z-axis and x-axis are $M_{1}$ and $M_{2}$, respectively. Suppose the array under consideration is uniform, with *d* denoting the channel spacing.

The radar transmits a burst of pulses at a constant pulse repetition frequency (PRF) during the flight, and, meanwhile, the antenna beam uniformly sweeps the detecting scenario in azimuth.

In forward-looking imaging, the signal emitted by the radar is the Linear Frequency Modulation (LFM) signal, which can be expressed as:(1)$$s_{\mathrm{LFM}}\left(\tau \right)=\mathrm{rect}\left(\frac{\tau }{T_{r}}\right)\exp\left(j\pi k_{r}\tau^{2}+j2\pi f_{c}\tau \right),$$
where τ represents the fast time, $T_{r}$ represents the pulse width, $k_{r}$ represents the linear modulation frequency, $f_{c}$ is the carrier frequency after the original signal modulation, $\mathrm{rect}\left(\cdot \right)$ is the rectangular window function.

Firstly, we construct the echo model of point P in Figure 1. Suppose its spherical coordinates are $\left(R_{a},\theta_{azi},\varphi_{pit}\right)$, the echo received by the radar is:(2)$$\begin{array}{cc} s_{e}\left(\tau ,t_{s}\right) & =A_{\mathrm{rcs}}h\left(t_{s}-\frac{\theta_{azi}}{\omega }\right)\mathrm{rect}\left[\frac{\tau -\frac{2R\left(t_{s}\right)}{c}-\tau_{a}}{T_{r}}\right]\\ & \exp\left\{j\pi k_{r}{\left[\tau -\frac{2R\left(t_{s}\right)}{c}-\tau_{a}\right]}^{2}-j2\pi f_{c}\left[\frac{2R\left(t_{s}\right)}{c}+\tau_{a}\right]\right\} \end{array},$$
where $s_{e}\left(\cdot \right)$ is the echo matrix of magnitude, $t_{s}$ represents the slow time of azimuth scanning, $A_{\mathrm{rcs}}$ is the amplitude value of the backscattering coefficient, $h\left(\cdot \right)$ represents the two-dimensional antenna pattern, ω is the angular velocity of a single scan, $\tau_{a}$ is the delay difference generated when the target echo reaches different receiving channels in the array, $R\left(t_{s}\right)$ is the range equation of radar.

According to geometry, $R\left(t_{s}\right)$ can be expressed as
(3)$$R\left(t_{s}\right)=\sqrt{R_{a}^{2}+{\left(v_{0}\cdot t_{s}\right)}^{2}-2R_{a}v_{0}t_{s}\cos\theta_{azi}\cos\varphi_{pit}},$$
where $v_{0}$ is the flight speed of the radar platform.

The Taylor expansion of the above formula is
(4)$$R(t_{s})=R_{a}-v_{0}t_{s}\cos\theta_{azi}\cos\varphi_{pit}+\frac{{v_{0}}^{2}(1-\cos^{2}\theta_{azi}\cos^{2}\varphi_{pit})}{2R_{0}}{t_{s}}^{2}+O({t_{s}}^{2}).$$

In forward-looking imaging, the angle range of imaging is narrow, and the radar scanning speed is extremely fast. At the same time, due to the large distance between the radar and P, the second-order and higher-order terms of Taylor’s expansion are negligible, and $\cos\theta_{azi}$ can be approximated to 1. So $R(t_{s})$ can be written as
(5)$$R(t_{s})=R_{a}-v_{0}t_{s}\cos\varphi_{pit}.$$

For $\cos\varphi_{pit}$, let us assume that the beamwidth of the pitched antenna is $\varphi_{0}$. Then, the range of $\varphi_{pit}$ is $\varphi_{n}\pm \varphi_{0}$. $\varphi_{n}$ is the pitch angle of the current beam. Because $\varphi_{0}$ is much smaller than $\varphi_{n}$, $\cos\varphi_{pit}$ is approximately written as $\cos\varphi_{n}$. So $R(t_{s})$ can be written as
(6)$$R\left(t_{s}\right)=R_{a}-v_{0}t_{s}\cos\varphi_{n}.$$

After pulse compression and range cell migration correction, the echo signal can be expressed as
(7)$$\begin{array}{ll} s_{e}\left(\tau ,t_{s}\right)= & A_{\mathrm{rcs}}h\left(t_{s}-\frac{\theta_{azi}}{\omega }\right)\sin c\left[B\left(\tau -\tau_{\mathrm{array}}-\frac{2R_{p}}{c}\right)\right]\exp\left(-j2\pi f_{c}\tau_{\mathrm{array}}\right)\cdot \\ & \exp\left[-j\frac{4\pi }{\lambda }R\left(t_{s}\right)\right]\; \end{array},$$
where $\lambda =c/f_{c}$ represents the wavelength of the carrier frequency and $B=k_{r}T_{r}$ is the bandwidth.

Selecting the array element (1,1) in Figure 1 as the reference element, so the delay difference $\tau_{a}\left(1,1\right)=0$, and the delay difference in the channel numbered $I_{m_{1}m_{2}}$ is $\tau_{a}\left(m_{1},m_{2}\right)$. It can be written as
(8)$$\tau_{a}\left(m_{1},m_{2}\right)=\frac{d}{c}\left[\left(m_{1}-1\right)\sin\theta_{azi}\cos\varphi_{pit}+\left(m_{2}-1\right)\sin\varphi_{pit}\right],$$
where d represents the channel spacing.

In this way, multi-channel echo data can be written as
(9)$$\begin{array}{l} s\left(\tau ,t_{s},m_{1},m_{2}\right)=A_{\mathrm{rcs}}h\left(t_{s}-\frac{\theta_{azi}}{\omega }\right)\exp\left[-j2\pi f_{c}\tau_{a}\left(m_{1},m_{2}\right)\right]\exp\left[-j\frac{4\pi }{\lambda }R\left(t_{s}\right)\right]\cdot \\ \;\;\;\;\;\;\;\;\;\;\;\;\;\;\;\;\;\;\;\;\sin c\left\{B\left[\tau -\tau_{a}\left(m_{1},m_{2}\right)-\frac{2R_{a}}{c}\right]\right\} \end{array}$$

After receiving the echo, we can derive the four-dimensional echo data, which consists of range, pulse, and azimuth-pitch 2D array. For a fixed-range cell, the azimuth-pitch model in a single pulse is constructed:

Selecting the channel (1,1) in Figure 1 for reference, the phase difference in the (m1, m2)th channel with respect to the reference channel is
(10)$$\Delta \phi_{m_{1}m_{2}}=-2\pi \frac{d}{\lambda }\left(\left(m_{1}-1\right)\sin\theta_{azi}\cos\varphi_{pit}\;\left.\;\;+\left(m_{2}-1\right)\sin\varphi_{pit}\right)\right.,$$
where $m_{1}=1,2,\cdots ,M_{1}$ and $m_{2}=1,2,\cdots ,M_{2}$. Let
(11)$$\left\{\begin{array}{l} u_{az}=2\pi d\sin\theta_{azi}\cos\varphi_{pit}/\lambda \\ v_{az}=2\pi d\sin\varphi_{pit}/\lambda \end{array}\right..$$

The steering vector of the array in pitch direction can be written as
(12)$$a_{z}\left(\theta_{azi},\varphi_{azi}\right)={\left[1,e^{-ju_{az}},\cdots ,e^{-j(M_{1}-1)u_{az}}\right]}^{T}.$$

The steering vector of the array in azimuth direction can be written as
(13)$$a_{x}\left(\theta_{azi},\varphi_{pit}\right)={\left[1,e^{-jv_{az}},\cdots ,e^{-j(M_{2}-1)v_{az}}\right]}^{T}.$$

The steering vector corresponding to the Pθi,φj can be written as
(14)aθazi,φpit=axθazi,φpit⊗azθazi,φpit,
where ⊗ denotes Kronecker product and aθazi,φpit∈ℂ  M1M2×1.

Considering the presence of additive white Gaussian noise, the array output after receiving echoes can be expressed by
(15)$$y(n)=a\left(\theta_{azi},\varphi_{pit}\right)\sigma_{az}(n)+e(n),$$
where y(n) denotes array output. n=1,2⋯,L is the index of the snapshots.$\sigma_{az}(n)$ is the backscatter coefficient of P received by the nth snapshot and $e\left(n\right)\in^{\;\;M_{1}M_{2}\times 1}$ is the interference noise.

Then, the echo model of the whole forward-looking region is constructed. As shown in Figure 2, the detection region in a range cell is divided into a series of 2D grids, where the azimuth and pitch direction are equally divided into $N_{1}$ and $N_{2}$ parts, respectively. Each grid corresponds to an individual $\theta_{i}$ and $\varphi_{j}$, where $1\leq i\leq N_{1}$ and $1\leq j\leq N_{2}$ [16]. Suppose there is a potential source in each grid. The information of each potential source needs to be estimated. Searching each 2D grid, the total steering vector matrix is given as
(16)$$A=\left[a\left(\theta_{1},\varphi_{1}\right),a\left(\theta_{1},\varphi_{2}\right),\cdots ,a\left(\theta_{N_{1}},\varphi_{N_{2}}\right)\right]$$
whose size is $M_{1}M_{2}\times N_{1}N_{2}$. Subsequently, the array output is expressed as
(17)y(n)=Aσ(n)+e(n),
where $\sigma (n)={\left(\sigma_{11}(n),\sigma_{12}(n),\cdots ,\sigma_{N_{1}N_{2}}(n)\right)}^{T}$.

## 3. Processing Algorithm

### 3.1. Two-Dimensional Iterative Adaptive Approach

IAA is a spectral estimation algorithm based on the weighted least squares (WLS) criterion [17]. It estimates the source distribution by multiple iterations instead of multiple snapshot data. This paper applies IAA to the processing of 2D spatial data.

Firstly, according to the WLS criterion, the error function of (17) is constructed:(18)$$\begin{array}{l} E={\left(y(n)-A\sigma (n)\right)}^{H}W\left(y(n)-A\sigma (n)\right)\\ \;\;\;=y^{H}Wy-y^{H}WA\sigma -\sigma^{H}A^{H}Wy\\ \;\;\;\;\;\;\;\;+\sigma^{H}A^{H}WA\sigma \end{array},$$
where W denotes weight matrix. Minimize the E with respect to σ(n):(19)$$\frac{dE}{d\sigma (n)}=-2A^{H}Wy+2A^{H}WA\sigma =0.$$

The WLS estimate of σ(n) can be determined as:(20)$$\hat{\sigma }(n)={\left(A^{H}WA\right)}^{-1}A^{H}Wy(n).$$

Since the huge computation of the ${\left(A^{H}WA\right)}^{-1}$, we estimate σ(n) for each grid individually. Considering the potential source at grid $\left(\theta_{i},\varphi_{j}\right)$, Equation (20) can be rewritten as:(21)$$\begin{array}{l} {\hat{\sigma }}_{ij}(n)={\left[a{\left(\theta_{i},\varphi_{j}\right)}^{H}Wa\left(\theta_{i},\varphi_{j}\right)\right]}^{-1}a^{H}\left(\theta_{i},\varphi_{i}\right)Wy(n)\\ \;\;\;\;\;\;\;\;\;\;\;\;\;\;=\;\;\;\frac{a^{H}\left(\theta_{i},\varphi_{i}\right)Wy(n)}{a^{H}\left(\theta_{i},\varphi_{j}\right)Wa\left(\theta_{i},\varphi_{j}\right)} \end{array}.$$

Then, determine the weight matrix W. Define the power matrix P which satisfies
(22)$$P=diag\left(p_{11},p_{12},\cdots ,p_{ij},\cdots ,p_{N_{1}N_{2}}\right),$$
where $p_{ij}$ is the power of the source located in the grid $\left(\theta_{i},\varphi_{j}\right)$. When (18) derives the minimum value, do a detailed derivation of W. The size of W is $M_{1}M_{2}\times M_{1}M_{2}$, so W can be written as
(23)$$W=\left[\begin{array}{ccc} w_{1} & \cdots & 0\\ \vdots & \ddots & \vdots \\ 0 & \cdots & w_{M_{1}M_{2}} \end{array}\right].$$

Suppose the variance of the error vector e=y(n)−Aσ(n) is
(24)$$\mathrm{var}(e)=\vartheta^{2}V,$$
where V is a positive definite matrix and can be written as
(25)$$V=TT^{H}.$$

Left-multiply the error vector e by $T^{-1}$:(26)$$T^{-1}e=T^{-1}y(n)-T^{-1}A\sigma (n).$$

Then, the variance of the new error function $\delta =T^{-1}e$ satisfies $\mathrm{var}(\delta )=\vartheta^{2}I$. In this case, the least square estimation results are optimal. According to the least squares criterion, $\hat{\sigma }(n)$ can be written as
(27)$$\begin{array}{l} \hat{\sigma }(n)={\left[{\left(T^{-1}A\right)}^{H}T^{-1}A\right]}^{-1}T^{-1}A^{H}y(n)\\ \;\;\;\;\;\;\;\;=\left(A^{H}T^{-H}T^{-1}A\right)A^{H}T^{-H}T^{-1}y(n)\\ \;\;\;\;\;\;\;\;={\left(A^{H}V^{-1}A\right)}^{-1}A^{H}V^{-1}s \end{array}.$$

Compare the above formula with (20), which is exactly equivalent, so
(28)$$W=V^{-1}.$$

$V\left(\theta_{i},\varphi_{j}\right)$ is the covariance matrix of clutter and noise [18]. It can be written as
(29)$$V\left(\theta_{i},\varphi_{j}\right)=R_{M_{1}M_{2}}-p_{ij}a\left(\theta_{i},\varphi_{j}\right)a^{H}\left(\theta_{i},\varphi_{j}\right),$$
where $R_{M_{1}M_{2}}\in^{\;\;M_{1}M_{2}\times M_{1}M_{2}}$ is the covariance matrix of the echo signal, which can be written as
(30)$$R_{M_{1}M_{2}}=APA^{H}.$$

Substitute (28) into (21); (21) can be rewritten as
(31)$${\hat{\sigma }}_{ij}(n)=\frac{a^{H}\left(\theta_{i},\varphi_{i}\right)V^{-1}\left(\theta_{i},\varphi_{j}\right)y(n)}{a^{H}\left(\theta_{i},\varphi_{j}\right)V^{-1}\left(\theta_{i},\varphi_{j}\right)a\left(\theta_{i},\varphi_{j}\right)}\;.$$

Since each ${\hat{\sigma }}_{ij}$ needs to calculate a different $V^{-1}\left(\theta_{i},\varphi_{j}\right)$, the computation in each iteration is huge. Applying the matrix inversion lemma [19], $V^{-1}\left(\theta_{i},\varphi_{j}\right)$ can be replaced by $R_{M_{1}M_{2}}^{-1}$. Equation (31) can be rewritten as
(32)$${\hat{\sigma }}_{ij}(n)=\frac{a^{H}\left(\theta_{i},\varphi_{i}\right)R_{M_{1}M_{2}}^{-1}y(n)}{a^{H}\left(\theta_{i},\varphi_{j}\right)R_{M_{1}M_{2}}^{-1}a\left(\theta_{i},\varphi_{j}\right)}\;.$$

Finally, the azimuth-pitch information is reconstructed more accurately by adaptive iteration. The iterative process is as follows: Let
(33)$$P=\left|\sigma (n)*conj\left(\sigma (n)\right)\right|,$$
where σ is the estimated value of the previous iteration and conj() denotes complex conjugation. According to (30) and (32), $R_{M_{1}M_{2}}$ and ${\hat{\sigma }}_{ij}$ are updated sequentially. The simulation results show that convergence can be realized in about five iterations with eight array elements. Notably, in the first iteration, P can be initialized by the following expression:(34)$${\hat{p}}_{ij}=\frac{\left|a^{H}\left(\theta_{i},\varphi_{j}\right)y(n)\right|}{a^{H}\left(\theta_{i},\varphi_{j}\right)a\left(\theta_{i},\varphi_{j}\right)}.$$

With the 2D-IAA processing, the azimuth-pitch super-resolution spectrum in a single range-pulse cell is obtained.

### 3.2. Three-Dimensional Forward-Looking Imaging Algorithm

The proposed algorithm process is as follows.

Firstly, the range super-resolution is realized by pulse compression.

Subsequently, the azimuth-pitch model is constructed. Applying 2D-IAA to process 2D spatial snapshot data of each range-pulse cell, a series of 2D super-resolution spectra is obtained.

Thirdly, the 2D super-resolution spectrum in the same range cell is accumulated by the following processing: In a range cell, the 2D super-resolution spectrum of each pulse corresponds to a unique beam center. According to these beam centers, each super-resolution spectrum is shifted to the angle corresponding to its beam center, which can be expressed as
(35)$$C_{r\_NEW}\left(\theta_{i},\varphi_{j}\right){=C}_{r\_OLD}\left(\theta_{i},\varphi_{j}\right)+\left|\hat{\sigma }(\theta_{i},\varphi_{j})\right|,$$
where $C_{r\_NEW}\left(\theta_{i},\varphi_{j}\right)$ and $C_{r\_OLD}\left(\theta_{i},\varphi_{j}\right)$ denote the image before and after the accumulation, ${\bar{\theta }}_{m}-\theta_{b}<\theta_{i}<{\bar{\theta }}_{m}+\theta_{b}$, $\varphi_{n}-\varphi_{0}<\varphi_{j}<\varphi_{n}+\varphi$, ${\bar{\theta }}_{m}$ denotes the instantaneous boresight direction of beam in azimuth at the nth pulse. $\theta_{b}$ denotes the boundary of azimuth angle, r represents the index of the range cell. where $\hat{\sigma }(\theta_{i},\varphi_{j})$ is the backscatter coefficient of the source at $\left(\theta_{i},\varphi_{j}\right)$.

Then, the azimuth-pitch 2D super-resolution spectrum in this range cell can be obtained.

Finally, after the above processing, the range-azimuth-pitch 3D super-resolution data in the polar coordinates system can be obtained. However, the array radar is used to observe targets located in the space rectangular coordinate system. It is inappropriate to image in the polar coordinates system. With the coordinate transformation, the 3D super-resolution data in the polar coordinates system are transformed into the space rectangular coordinate system. Combined with 3D mapping, the 3D forward-looking super-resolution imaging is realized. The algorithm flow chart is shown in Figure 3.

## 4. Simulation Result

In order to verify the validity of the proposed method, the point targets simulation, and scene simulation of both are given, respectively.

### 4.1. Point Targets Simulation

The distribution of the five-point targets is shown in Figure 4. The parameters of the missile-borne array radar are given in Table 1 (Value 1). Figure 5 illustrates the 3D forward-looking imaging results in the space rectangular coordinate system. Figure 5a is the imaging result based on the real beam method. It relies on matched filtering to improve azimuth-pitch resolution. However, due to the presence of the sidelobe in the antenna pattern, the imaging result in the azimuth-pitch direction has distinct sidelobes, which causes the generation of false targets. The sidelobe can be suppressed by adding a window function, but the imaging resolution remains poor. The targets in a range profile develop aliasing, although the position information about the targets realizes a rough reconstruction. Figure 5b is the imaging result based on 2D-IAA. It can be seen that all point targets are precisely reconstructed, and the adjacent targets in a range profile can be distinctly distinguished.

Subsequently, select the range profile of R=1000 m to observe the azimuth-pitch resolution. The imaging results using the real beam method and 2D-IAA are shown in Figure 6a and Figure 6b, respectively. Obviously, the imaging result using 2D-IAA has a better resolution.

### 4.2. Scene Simulation

The imaging scene of simulated ship targets is given in Figure 7. We focus on the scene reconstruction of the ship and the jammers in Figure 7. In addition, the major radar parameters are given in Table 1 (Value 2).

Figure 8 shows the 3D imaging results of the scene. Figure 8a is the 3D imaging result based on the real beam method. Figure 8b is the 3D imaging result based on 2D Orthogonal Matching Pursuit (2D-OMP) [20]. In Figure 8a, it can be seen that the ship and the jammers are overlapped, and the imaging resolution is low in the x-z plane. Figure 8c shows the 3D imaging results based on the proposed method. Obviously, the jammers and the ship can be clearly distinguished. By comparing Figure 8a–c, the validity of the proposed method can be proved.

Figure 9 shows the x-y plane projections of the 3D imaging. It can be seen that the 2D forward-looking imaging results can also be obtained by the projection of 3D imaging results in the x-y plane. Furthermore, the imaging result of Figure 9c has a better resolution in the x-direction than the imaging result of Figure 9a,b.

Limited by the mapping method, the intensity of targets cannot be directly observed. Therefore, we select the section of x=−25 m to observe the intensity of different targets. Take the jammer at y=1040 m as an example. In Figure 10a, the imaging result based on the real beam method is not accurate in estimating the amplitude of this jammer, resulting in a large deviation and unsatisfactory imaging effect. In Figure 10b, according to the imaging results obtained by the method based on 2D-OMP, the amplitude estimation of the jammer is relatively accurate, and the imaging resolution and positioning accuracy are also improved, but good focusing still cannot be achieved. In Figure 10c, through the imaging method proposed in this paper, the amplitude information of interference can be accurately restored, better focusing can be achieved, and the reconstructed target position information is more consistent with the real target. Therefore, the performance of the proposed method is further verified.

## 5. Conclusions

In order to improve the 3D forward-looking imaging ability of missile borne radar system, this paper proposes a 3D imaging algorithm based on 2D-IAA processing. This algorithm first applies 2D-IAA to the data received by a planar array, making the azimuth-pitch information of the target to be imaged be accurately reconstructed with a single snapshot. Then, the 2D super-resolution spectrum in the same range cell is accumulated according to their beam centers, and this operation is repeated for each range cell. Through the imaging simulation of the point targets and the ship targets on the sea, the validity of the proposed algorithm is demonstrated. Compared with the 3D imaging algorithm based on the real beam method and that based on 2D-OMP, this algorithm gains better imaging performance.

## Figures and Tables

**Figure 1 sensors-24-07356-f001:**
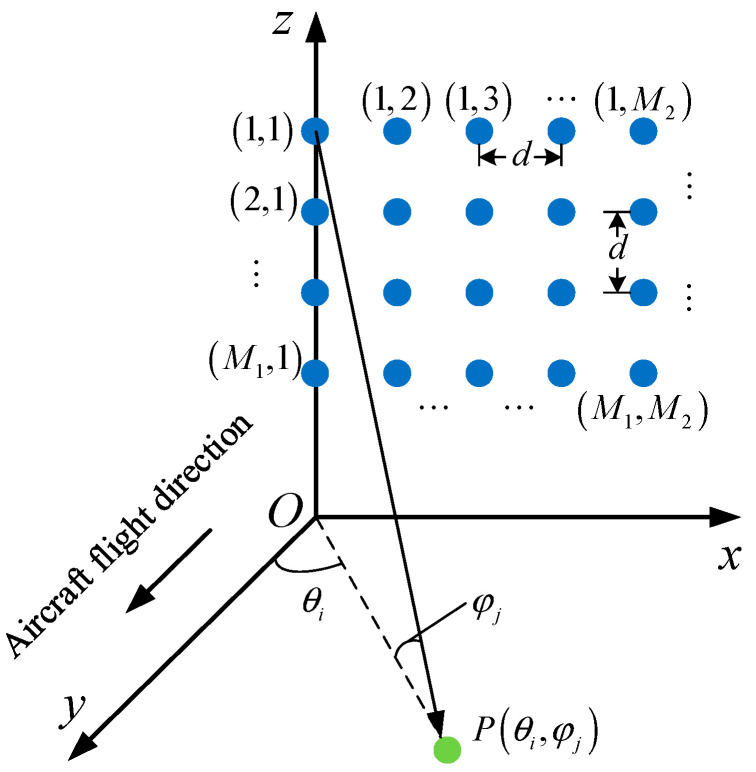
Distribution of the array elements.

**Figure 2 sensors-24-07356-f002:**
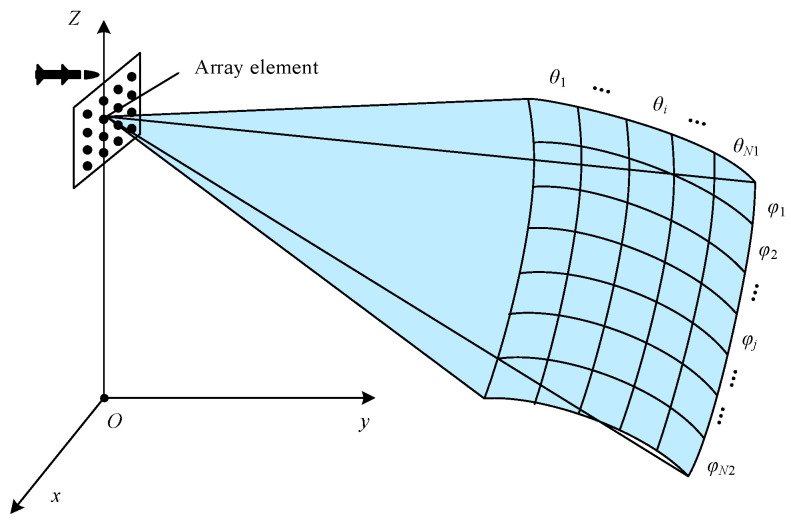
Angular division of the observed region.

**Figure 3 sensors-24-07356-f003:**
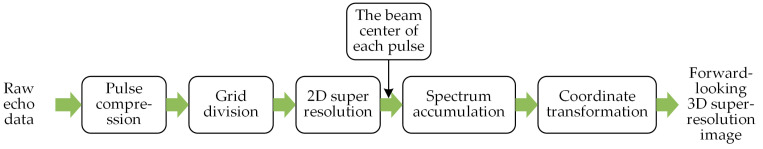
Signal processing flow.

**Figure 4 sensors-24-07356-f004:**
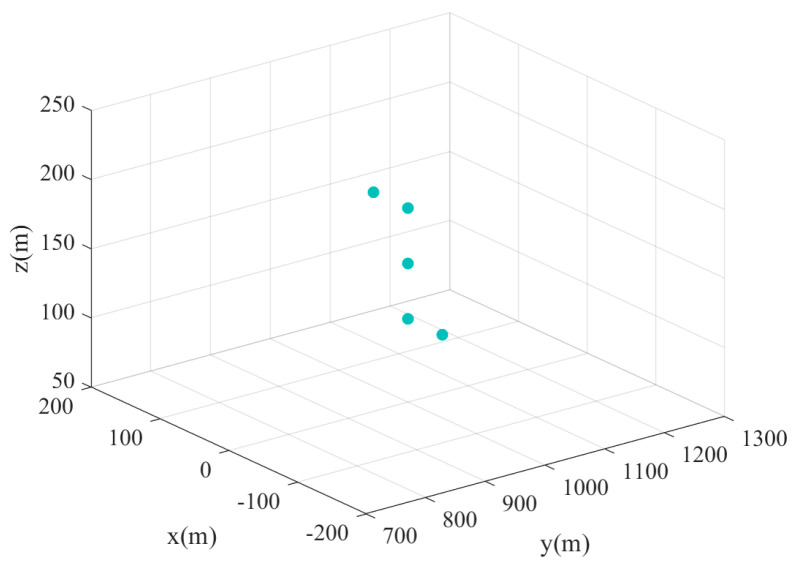
Point targets distribution.

**Figure 5 sensors-24-07356-f005:**
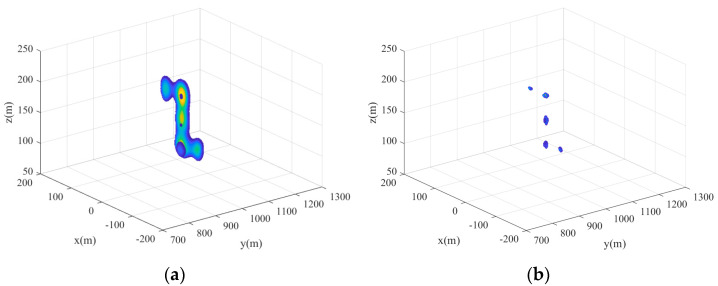
Simulation results of the point targets. (**a**) Real beam method. (**b**) 2D-IAA.

**Figure 6 sensors-24-07356-f006:**
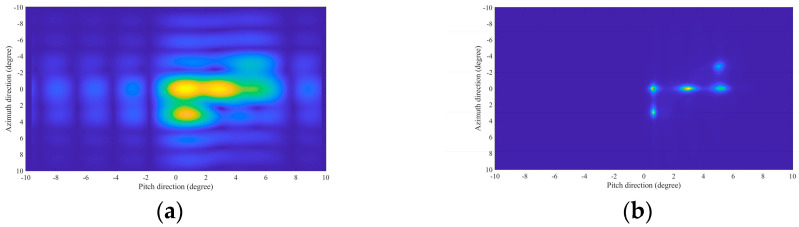
Simulation results in the range profile of R=1000 m. (**a**) Real beam method. (**b**) 2D-IAA.

**Figure 7 sensors-24-07356-f007:**
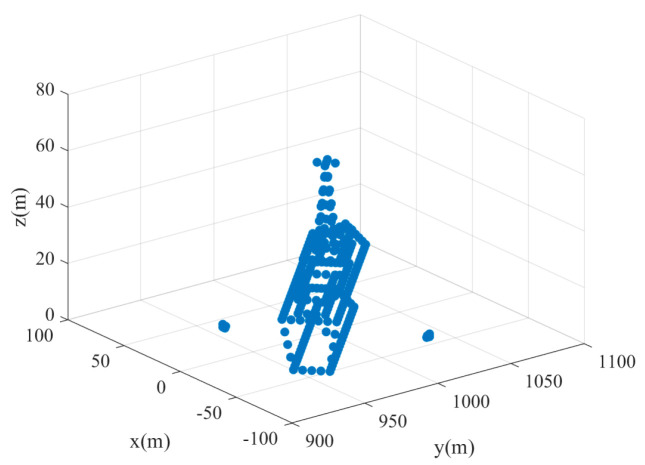
Scene for ship target imaging simulation.

**Figure 8 sensors-24-07356-f008:**
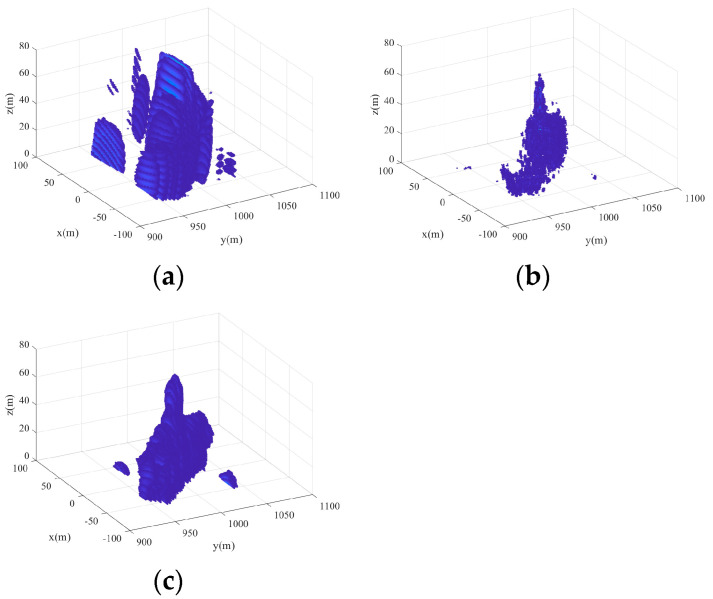
The 3D imaging results of the scene. (**a**) Real beam method. (**b**) 2D-OMP. (**c**) 2D-IAA.

**Figure 9 sensors-24-07356-f009:**
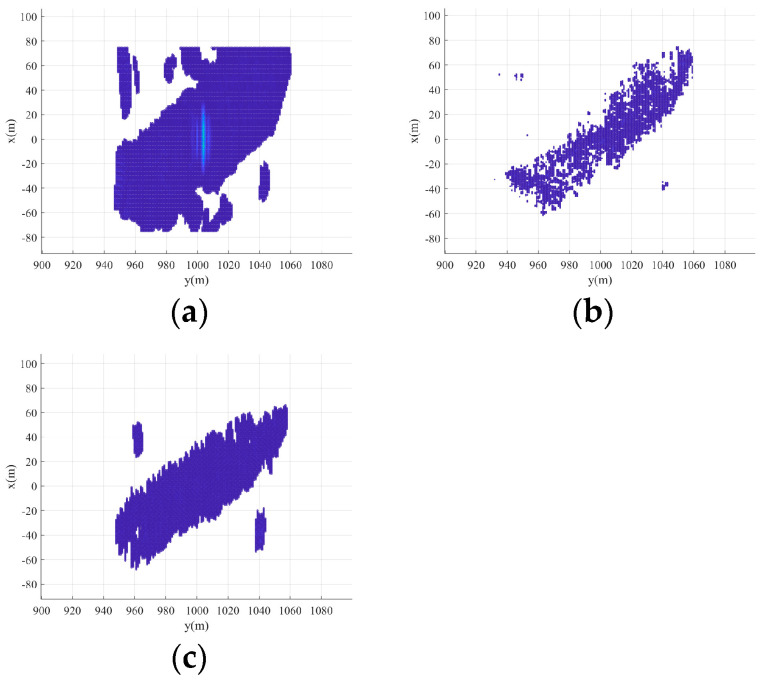
The projection of the 3D imaging results in x-y plane. (**a**) Real beam method. (**b**) 2D-OMP. (**c**) 2D-IAA.

**Figure 10 sensors-24-07356-f010:**
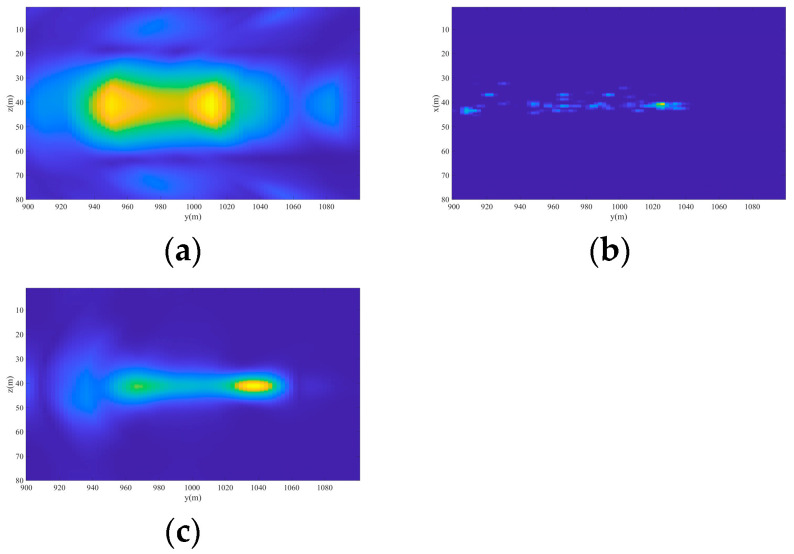
The section of the 3D imaging results at x=−25 m. (**a**) Real beam method. (**b**) 2D-OMP. (**c**) 2D-IAA.

**Table 1 sensors-24-07356-t001:** RADAR PARAMETER.

Parameter	Value 1	Value 2
Carrier frequency	18 GHz	18 GHz
Beam scanning velocity	150°/s	150°/s
PRF	1000 Hz	3000 Hz
Bandwidth	125 MHz	125 MHz
Platform velocity	750 m/s	750 m/s
Number of receiving channels	8 × 8	8 × 8
Channel spacing	0.05 m	0.05 m
Radar altitude	200 m	200 m

## Data Availability

Data are contained within the article.

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
