# Peer review of "Array Radar Three-Dimensional Forward-Looking Imaging Algorithm Based on Two-Dimensional Super-Resolutionâ€"

_sensors, 2024, doi:10.3390/s24227356_

Round 1
Reviewer 1 Report
Comments and Suggestions for Authors
1. The duplication of the manuscript with published conference paper titled “A Three-Dimensional Forward-Looking Imaging Algorithm Based on 2D Iterative Adaptive Approach” is unacceptable, so the manuscript must be rewritten in a different form with emphasis on the developments from this conference paper.
2. The section of Three-Dimensional Forward-Looking Imaging Algorithm needs to be presented in more detail to be able to show the relationship between the signal components received from the antenna array as well as the displacement of the antenna array in the system signal model with the processing in this algorithm. For example, it is necessary to represent the snapshots with an index and show that index in the explanation for the algorithm in this section.
3. The authors please explain clearly the relationship between parameter d=lambda/2 in model description at line 93 with "Array element spacing" in table 1; and the relationship between Platform velocity parameter in table 1 and the algorithm parameters/variables?
4. The authors please explain clearly the differences between the scene of sea surface targets and the point targets. What features of the sea surface model were used in this simulation?
5. There is a conflict in the lines 50-52 and 53.
6. The 2D Orthogonal Matching Pursuit (2D-OMP) algorithm at lines 221-222 should be introduced or cited.
No Comment.
Author Response
Comments 1:[The duplication of the manuscript with published conference paper titled “A Three-Dimensional Forward-Looking Imaging Algorithm Based on 2D Iterative Adaptive Approach” is unacceptable, so the manuscript must be rewritten in a different form with emphasis on the developments from this conference paper.]
Response 1:[According to this comment, we have reorganized and revised parts of Section II and Section III to better describe the algorithm of this manuscript. (see line 99-105, 115, 117 and 127-129).
We completed this manuscript based on part of the research work in the above-mentioned conference paper (related explanation is also provided in the revise manuscript in line 282, according to this comment).
First of all, the previous work mainly focused on the three-dimensional imaging of airborne radar, but this manuscript is based on the missile-borne radar platform. Missile-borne radar is an important part of missile system, which provides accurate target guidance and detection and locking capability. However, current radar imaging methods don’t have the ability of high-resolution 3D forward-looking imaging. To remedy this, the research of this paper mainly focuses on the 3D forward-looking imaging for missile-borne radar.
Secondly, under the parameter conditions of the missile-borne platform, we increased the flight speed of the radar platform in the simulation experiment, as well as the distance to the scene center, and re-simulate the point target imaging to verify whether the method proposed in this paper is also effective in the application scenario of the missile-borne radar. At the same time, in order to further verify the effectiveness of the method, we also added the simulation of ship target imaging, and set the simulation scene as the ship and its adjacent jammers.
At last, in the conference paper, we only compared the imaging results of the proposed method with those based on the real beam method. In this paper, we added comparison and analysis with the imaging results based on two-dimension Orthogonal Matching Pursuit (2D-OMP). By comparing the imaging results of the scene obtained based on these three imaging methods, the validity of the proposed method was proved. By comparing the x-y plane projections of 3D imaging, we proved that using the proposed method can obtain better imaging results in the x-direction. Last but not the least, we selected the section of to observe the intensity of different targets. The results show that the proposed method can better recover the intensity information about the target.
The above-mentioned work is the expansion of the conference paper, which fully shows that the proposed method maintains the effectiveness of imaging capability of various targets under the background of missile-borne radar.]
Comments 2:[The section of Three-Dimensional Forward-Looking Imaging Algorithm needs to be presented in more detail to be able to show the relationship between the signal components received from the antenna array as well as the displacement of the antenna array in the system signal model with the processing in this algorithm. For example, it is necessary to represent the snapshots with an index and show that index in the explanation for the algorithm in this section.]
Response 2:[Thank you very much for your suggestion. We rewrote some statements in the part of model building. We also added the description of the algorithm and modified the equations, making the whole paper more clear and easier to understand (see line 99-105 and 127-129). Besides, we have used the index to represent the snapshot according to your request in equation (6), (8)-(13), (19), (20).]
Comments 3:[The authors please explain clearly the relationship between parameter d=lambda/2 in model description at line 93 with "Array element spacing" in table 1; and the relationship between Platform velocity parameter in table 1 and the algorithm parameters/variables?]
Response 3:[In practical processing of array radar, echo signals are often received in a multi-channel (sub-array) model. Thus, the variable d is actually employed to represent the spacing of the independent receiving channels of the array. So, the limitation of is deleted in the revised manuscript (see line 99-103). Thank the reviewer for pointing out this problem. We have also made the relevant changes in Table.1 of the manuscript to avoid confusion (see Table.1).
The platform velocity parameter is mainly used for the early echo generation and distance pulse compression processing, while the algorithm proposed in this paper is mainly used for the subsequent orientation and pitch orientation information processing, so it is not reflected in the paper.]
Comments 4:[The authors please explain clearly the differences between the scene of sea surface targets and the point targets. What features of the sea surface model were used in this simulation?]
Response 4:[In fact, in this paper, the echo model of a ship is constructed by using a group of point-targets, and two simulated passive jammers (corner reflector) are simulated in the adjacent area. Based on this comment, we made changes in the simulation section (see line 257)]
Comments 5:[There is a conflict in the lines 50-52 and 53.]
Response 5:[Thank you very much for your comments on the writing of the manuscript. Indeed, as you said, there are some contradictions in the semantics before and after, which has been modified in the manuscript and marked in red font (see line 53).]
Comments 6:[The 2D Orthogonal Matching Pursuit (2D-OMP) algorithm at lines 221-222 should be introduced or cited.]
Response 6:[References related to OMP algorithm have been added to the manuscript, please see line 335, in the revised version of this manuscript.]
Reviewer 2 Report
Comments and Suggestions for Authors
This paper proposes a 3D forward-looking imaging algorithm for missile-borne radar, and a 2D super-resolution algorithm is given to improve the resolution of 3D imaging. Overall the paper illustrates the idea clearly, and the results are satisfied. It is suggested that the author could consider the following comments for the publication of the paper.
1. The paper format needs to be more standardized in terms of the parameters and Equations, for example, the equation number should be flushed right and the equation location should be in the middle of the row.
2. Some recent super-resolution algorithms can be refered with analysis in the introduction, as they provide considerable methods on obtaining 2D super-resolution spectra, and may leads to new directions of current work, e.g.
[R1] Global and Local Context-Aware Ship Detector for High-Resolution SAR Images, IEEE Transactions on Aerospace and Electronic Systems, 2023
[R2] Enhanced Doppler Resolution and Sidelobe Suppression Performance for Golay Complementary Waveforms, Remote Sensing, 2023
[R3] Delay-Doppler Map Shaping through Oversampled Complementary Sets for High-Speed Target Detection, Remote Sensing, 2024
3. Is there any difference on the five points in Section 3.1? Please explain the reason where there are significant energy differences between each targets.
4. Section 3.2 can be summarized in a Table.
5. Why only PRF changes in two simulations?
6. The illustration of Section 4.2 is too simple, which needs to increase sufficient quantitative results.
Author Response
Comments 1:[ The paper format needs to be more standardized in terms of the parameters and Equations, for example, the equation number should be flushed right and the equation location should be in the middle of the row.]
Response 1:[Thank you very much for your pointing out of the format of this paper. We have revised the manuscript to meet the standard of Sensors. The modified parts in the new version of the manuscript are marked in red font.]
Comments 2:[Some recent super-resolution algorithms can be refered with analysis in the introduction, as they provide considerable methods on obtaining 2D super-resolution spectra, and may leads to new directions of current work, e.g.
[R1] Global and Local Context-Aware Ship Detector for High-Resolution SAR Images, IEEE Transactions on Aerospace and Electronic Systems, 2023
[R2] Enhanced Doppler Resolution and Sidelobe Suppression Performance for Golay Complementary Waveforms, Remote Sensing, 2023
[R3] Delay-Doppler Map Shaping through Oversampled Complementary Sets for High-Speed Target Detection, Remote Sensing, 2024]
Response 2:[Thank you very much for your recommendation and suggestion. Reference [R2] and [R3] are added in the reference list of this paper (see line 314-318)., and related description are also provided in the introduction (see line 69-76). Your recommendation also provides new ideas and directions for our subsequent work.]
Comments 3:[Is there any difference on the five points in Section 3.1? Please explain the reason where there are significant energy differences between each targets.]
Response 3:[The reason for the significant energy difference of the five points is that during the simulation, the energy of the target is set as a random number, and the energy of each point is not the same. Moreover, the positions of the five points are not accurately distributed on the pre-divided angle grid, which also leads to the difference in the energy of the five targets in the final image.]
Comments 4:[Section 3.2 can be summarized in a Table.]
Response 4:[According to your suggestion, we have added the algorithm flow chart in Section 3.2 of the revised manuscript (see line 204).]
Comments 5:[ Why only PRF changes in two simulations?]
Response 5:[In the two simulations, the distances between the radar platform and the target are different. In the simulation of the point target, the distance between the radar platform and the target is longer, and it takes a longer time to wait for the return of the echo signal. In order to simulate the actual situation more truly, the PRF is set at a lower level.]
Comments 6:[ The illustration of Section 4.2 is too simple, which needs to increase sufficient quantitative results.]
Response 6:[Specific quantitative analysis and explanation have been added in lines 245-254 of the paper, please check.]
Round 2
Reviewer 1 Report
Comments and Suggestions for Authors
The authors presented the Array Radar Three-Dimensional Forward-Looking Imaging Algorithm Based on Two-Dimensional Super-Resolution. The manuscript presented Azimuth-Pitch echo model, algorithm, and demonstrated the simulation results based on 2D-IAA algorithm. The authors did make some changes based on the reviewer's previous comments.
However, there still have some issues:
- The manuscript still did not show a clear distinguish between this version and the previous conference paper. The speed of platform was not well mentioned and presented in the manuscript. The algorithm presented in this manuscript was simply as a repetition from the published conference paper without any significant improvements or modifications.
- In the abstract, from line 10 to 14, the authors did not make any change in this paragraph when compare with previous version.
- In the introduction, from line 69 to 76, the authors introduced some methods such as PTP and PMuP. Can you give the relationship between this method and the author’s method? Can you compare the efficiency of the chosen method and other methods?
- Please check the term “Equation (16)” In line 172.
- The section of “Three-Dimensional Forward-Looking Imaging Algorithm” needs to be presented in more detail with the relationship between the signal components received from the antenna array as well as the displacement of the antenna array in the system signal model with the processing in this algorithm. The additional algorithm flowchart is not enough. The authors should present this in formula or equation forms.
- With the two scenarios for point targets distribution in the conference paper and that in this manuscript, there was no improvement in spatial resolution. Please make comparison and explanation for this?
- From line 229 to 263, there were figure index mistakes. The authors need to correct the numbers of these figures.
With the shortages of the manuscript as the comments above, I recommend that the manuscript needs a major revise before it can be considered for publication.
Comments on the Quality of English LanguageThe presentation of the manuscript should be improved.
Reviewer 2 Report
Comments and Suggestions for Authors
The comments have all been replied. I have no more questions.
Author Response
Comments 1:The comments have all been replied. I have no more questions.
Response 1: Thank you for all your suggestions.
Round 3
Reviewer 1 Report
Comments and Suggestions for Authors
This manuscript version has demonstrated an effort of the authors in improvement of the paper presentation, the signal model was added and the algorithm has been presented in more detail. However, it needs some minor revises before it can be considered to accept to publish on Sensors.
- In line 101, there is a spelling mistake “Liner”.
- The arrangement of Figure 1 needs to be suitable to the mention in the text.
- Please recheck all the citation, it seems like there are some miscorrelations to the previous version, some references cited do not show the relevant to the content of the text. For instance:
+ In line 167, the authors should review the relevant of the statement with article [19].
+ In line 180 seem not to mention the one-dimensional array signal processing for source localization as the reference [15].
- For the scenario simulation, it is recommended the name “surface ship target” should be changed to other name, because in this scenario simulation the model is not represented by a surface of the ship, it is more likely a high-density distribution of dots that shapes the ship.
- The conclusion needs to be improved.
Comments on the Quality of English LanguageThere are some spelling mistakes.
